# Spatio-temporal point processes with deep non-stationary kernels

**Zheng Dong[†], Xiuyuan Cheng[⋆], Yao Xie[†]**
† H. Milton Stewart School of Industrial and Systems Engineering, Georgia Institute of Technology
⋆ Department of Mathematics, Duke University
`yao.xie@isye.gatech.edu`

## Abstract

Point process data are becoming ubiquitous in modern applications, such as social networks, health care, and finance. Despite the powerful expressiveness of the popular recurrent neural network (RNN) models for point process data, they may not successfully capture sophisticated non-stationary dependencies in the data due to their recurrent structures. Another popular type of deep model for point process data is based on representing the influence kernel (rather than the intensity function) by neural networks. We take the latter approach and develop a new deep non-stationary influence kernel that can model non-stationary spatio-temporal point processes. The main idea is to approximate the influence kernel with a novel and general low-rank decomposition, enabling efficient representation through deep neural networks and computational efficiency and better performance. We also take a new approach to maintain the non-negativity constraint of the conditional intensity by introducing a log-barrier penalty. We demonstrate our proposed method's good performance and computational efficiency compared with the state-of-the-art on simulated and real data.

## 1 Introduction

Point process data, consisting of sequential events with timestamps and associated information such as location or category, are ubiquitous in modern scientific fields and real-world applications. The distribution of events is of great scientific and practical interest, both for predicting new events and understanding the events' generative dynamics (Reinhart, 2018). To model such discrete events in continuous time and space, spatio-temporal point processes (STPPs) are widely used in a diverse range of domains, including modeling earthquakes (Ogata, 1988; 1998), the spread of infectious diseases (Schoenberg et al., 2019; Dong et al., 2021), and wildfire propagation (Hering et al., 2009).

A modeling challenge is to accurately capture the underlying generative model of event occurrence in general spatio-temporal point processes (STPP) while maintaining the model efficiency. Specific parametric forms of conditional intensity are proposed in seminal works of Hawkes process (Hawkes, 1971; Ogata, 1988) to tackle the issue of computational complexity in STPPs, which requires evaluating the complex multivariate integral in the likelihood function. They use an exponentially decaying *influence kernel* to measure the influence of a past event over time and assume the influence of all past events is positive and linearly additive. Despite computational simplicity (since the integral of the likelihood function is avoided), such a parametric form limits the model's practicality in modern applications.

Recent models use neural networks in modeling point processes to capture complicated event occurrences. RNN (Du et al., 2016) and LSTM (Mei and Eisner, 2017) have been used by taking advantage of their representation power and capability in capturing event temporal dependencies. However, the recurrent structures of RNN-based models cannot capture long-range dependency (Bengio et al., 1994) and attention-based structure (Zhang et al., 2020; Zuo et al., 2020) is introduced to address such limitations of RNN. Despite much development, existing models still cannot sufficiently capture spatio-temporal non-stationarity, which are common in real-world data (Graham et al., 2013; Dong et al., 2021). Moreover, while RNN-type models may produce strong prediction performance, the models consist of general forms of network layers and the modeling power relies on the hidden states, thus often not easily interpretable.

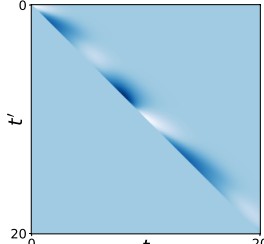
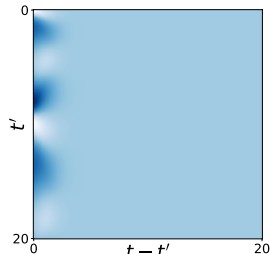

(a) Kernel matrix of $k(t', t)$ with rank 298.      (b) Kernel matrix of $k(t', t - t')$ with rank 7.

Figure 1: An example: equivalent representation of kernel by "displacement", from $k(t', t)$ to $k(t', t - t')$ can significantly decrease the rank of the kernel function: from 298 to 7, where $t'$ and $t$ represent the historical event time and the current time, respectively. On fitting the one-dimensional synthetic data generated by $k$, the model parameterized with $(t', t - t')$ outperforms the model parameterized with $(t', t)$. See Section 5 and Appendix C for experiment and formulation details.

A promising approach to overcome the above model restrictions is point process models that combine statistical models with neural network representation, such as Zhu et al. (2022) and Chen et al. (2020), to enjoy both the interpretability and expressive power of neural networks. In particular, the idea is to represent the (possibly non-stationary) influence kernel based on a spectral decomposition and represent the basis functions using neural networks. However, the prior work (Zhu et al., 2022) is not specifically designed for non-stationary kernel and the low-rank representation can be made significantly more efficient, which is the main focus of this paper.

**Contribution.** In this paper, we develop a non-stationary kernel (referred to as `DNSK`) for (possibly non-stationary) spatio-temporal processes that enjoy efficient low-rank representation, which leads to much improved computational efficiency and predictive performance. The construction is based on an interesting observation that by reparameterize the influence kernel from the original form of $k(t', t)$, (where $t'$ is the historical even time, and $t$ is the current time) to an equivalent form $k(t', t - t')$ (which thus is parameterized by the displacement $t - t'$ instead), the rank can be reduced significantly, as shown in Figure 1. This observation inspired us to design a much more efficient representation of the non-stationary point processes with much fewer basis functions to represent the same kernel.

In summary, the contributions of our paper include

- We introduce an efficient low-rank representation of the influence kernel based on a novel "displacement" re-parameterization. Our representation can well-approximate a large class of general non-stationary influence kernels and is generalizable to spatio-temporal kernels (also potentially to data with high-dimensional marks). Efficient representation leads to lower computational cost and better prediction power, as demonstrated in our experiments.

- In model fitting, we introduce a log-barrier penalty term in the objective function to ensure the non-negative conditional intensity function so the model is statistically meaningful, and the problem is numerically stable. This approach also enables the model to learn general influence functions (that can have negative values), which is a drastic improvement from existing influence kernel-based methods that require the kernel functions to be non-negative.

- Using extensive synthetic and real data experiments, we show the competitive performance of our proposed methods in both model recovery and event prediction compared with the state-of-the-art, such as the RNN-based and transformer-based models.

## 1.1 RELATED WORKS

The original work of A. Hawkes (Hawkes, 1971) provides classic self-exciting point processes for temporal events, which express the conditional intensity function with an influence kernel and a base rate. Ogata (1998) proposes a parametric form of spatio-temporal influence kernel which enjoys strong model interpretability and efficiency. However, such simple parametric forms own limited expressiveness in characterizing the complex events' dynamic in modern applications (Zhu et al., 2021; Liao et al., 2022).

Neural networks have been widely adopted in point processes (Xiao et al., 2017; Chen et al., 2020; Zhu et al., 2021). Du et al. (2016) incorporates recurrent neural networks and Mei and Eisner (2017) use a continuous-time invariant of LSTM to model event influence with exponential decay over time.

These RNN-based models may be unable to capture complicated event dependencies due to the recurrent structure. Zhang et al. (2020); Zuo et al. (2020) introduce self-attentive structures into point processes for their capability to memorize long-term influence by dealing with an event sequence as a whole. The main limitation is that they assume a dot-product-based score function and assume linearly decaying of event influence. Omi et al. (2019) propose a fully-connected neural network to model the cumulative intensity function to go beyond parametric decaying influence. However, the event embeddings are still generated by RNN, and fitting cumulative intensity function by neural networks lacks model interpretability. Note that all the above models tackle temporal events with categorical marks, which are inapplicable in continuous time and location space.

Recent works adopt neural networks in learning the influence kernel function. The kernel introduced in Okawa et al. (2021) uses neural networks to model the latent dynamic of time interval but still assumes an exponentially decaying influence over time. Zhu et al. (2022) proposes a kernel representation using spectral decomposition and represents feature functions using deep neural networks to harvest powerful model expressiveness when dealing with marked event data. Our method considers an alternative novel kernel representation that allows the general kernel to be expressed further low-rankly.

## 2    BACKGROUND

**Spatio-temporal point processes** (STPPs) (Reinhart, 2018; Moller and Waagepetersen, 2003) have been widely used to model sequences of random events that happen in continuous time and space. Let $\mathcal{H} = \{(t_i, s_i)\}_{i=1}^n$ denote the event stream with time $t_i \in [0, T] \subset \mathbb{R}$ and location $s_i \in \mathcal{S} \subset \mathbb{R}^{d_\mathcal{S}}$ of $i$th event. The event number $n$ is also random. Given the observed history $\mathcal{H}_t = \{(t_i, s_i) \in \mathcal{H} | t_i < t\}$ before time $t$, an STPP is then fully characterized by the conditional intensity function

$$\lambda(t, s \mid \mathcal{H}_t) = \lim_{\Delta t \downarrow 0, \Delta s \downarrow 0} \frac{\mathbb{E}\left[\mathbb{N}([t, t + \Delta t] \times B(s, \Delta s)) \mid \mathcal{H}_t\right]}{|B(s, \Delta s)|\Delta t}, \tag{1}$$

where $B(s, \Delta s)$ is a ball centered at $s \in \mathbb{R}^{d_\mathcal{S}}$ with radius $\Delta s$, and the counting measure $\mathbb{N}$ is defined as the number of events occurring in $[t, t + \Delta t] \times B(s, \Delta s) \subset \mathbb{R}^{d_\mathcal{S}+1}$. Naturally $\lambda(t, s | \mathcal{H}_t) \geq 0$ for any arbitrary $t$ and $s$. In the following, we omit the dependency on history $\mathcal{H}_t$ and use common shorthand $\lambda(t, s)$. The log-likelihood of observing $\mathcal{H}$ on $[0, T] \times \mathcal{S}$ is given by (Daley et al., 2003)

$$\ell(\mathcal{H}) = \sum_{i=1}^n \log \lambda(t_i, s_i) - \int_0^T \int_\mathcal{S} \lambda(t, s) ds dt \tag{2}$$

**Neural point processes** parameterize the conditional intensity function by taking advantage of recurrent neural networks (RNNs). In Du et al. (2016), an input vector $\boldsymbol{x}_i$ which extracts the information of event $t_i$ and the associated event attributes $m_i$ (can be event mark or location) is fed into the RNN. A hidden state vector $\boldsymbol{h}_i$ is updated by $\boldsymbol{h}_i = \rho(\boldsymbol{h}_{i-1}, \boldsymbol{x}_i)$, where $\rho(\cdot)$ is a mapping fulfilled by recurrent neural network operations. The conditional intensity function on $(t_i, t_{i+1}]$ is then defined as $\lambda(t) = \delta(t, \boldsymbol{h}_i)$, where $\delta$ is an exponential transformation that guarantees a positive intensity. In Mei and Eisner (2017) the RNN is replaced by a continuous-time LSTM module with hidden states $\boldsymbol{h}(t)$ defined on $[0, T]$ and a Softplus function $\delta$. Attention-based models are introduced in Zuo et al. (2020); Zhang et al. (2020) to overcome the inability of RNNs to capture sophisticated event dependencies due to their recurrent structures.

**Hawkes process** (Hawkes, 1971) is a well-known generalized point process model. Assuming the influences from past events are linearly additive, the conditional intensity function takes the form of

$$\lambda(t, s) = \mu + \sum_{(t', s') \in \mathcal{H}_t} k(t', t, s', s), \tag{3}$$

where $k$ is an influence kernel function that captures event interactions. Commonly the kernel function is assumed to be *stationary*, that is, $k$ only depends on $t - t'$ and $s - s'$, which limits the model expressivity. In this work, we aim to capture complicated non-stationarity in spatio-temporal event dependencies by leveraging the strong approximation power of neural networks in kernel fitting.

## 3    LOW-RANK DEEP NON-STATIONARY KERNEL

Due to the intricate dependencies between events, it is challenging to choose the form of kernel function that achieves great model expressiveness while enjoying high model efficiency. In this

section, we introduce a unified model with a low-rank deep non-stationary kernel to capture the complex heterogeneity in events' influence over spatio-temporal space.

### 3.1 KERNEL WITH HISTORY AND SPATIO-TEMPORAL DISPLACEMENT

For the influence kernel function $k(t', t, s', s)$, by using the displacements in $t$ and $s$ as variables, we first re-parameterize the kernel as $k(t', t-t', s', s-s')$, where the minus in $s-s'$ refers to element-wise difference between $s$ and $s'$ when $d_{\mathcal{S}} > 1$. Then we achieve a finite-rank decomposed representation based on (truncated) singular value decomposition (SVD) for kernel functions (Mollenhauer et al., 2020) (which can be understood as the kernel version of matrix SVD, where the eigendecomposition is based on Mercer's Theorem (Mercer, 1909)), and that the decomposed spatial (and temporal) kernel functions can be approximated under shared basis functions (cf. Assumption A.2). The resulting approximate finite-rank representation is written as (details are in Appendix A.1)

$$k(t', t - t', s', s - s') = \sum_{r=1}^{R} \sum_{l=1}^{L} \alpha_{lr} \psi_l(t') \varphi_l(t - t') u_r(s') v_r(s - s'). \tag{4}$$

Here $\{\psi_l, \varphi_l : [0, T] \to \mathbb{R}, l = 1, \dots, L\}$ are two sets of temporal basis functions that characterize the temporal influence of event at $t'$ and the decaying effect brought by elapsed time $t - t'$. Similarly, spatial basis functions $\{u_r, v_r : \mathcal{S} \to \mathbb{R}, r = 1, \dots, R\}$ capture the spatial influence of event at $s'$ and the decayed influence after spreading over the displacement of $s - s'$. The corresponding weight $\alpha_{lr}$ at different spatio-temporal ranks combines each set of basis functions into a weighted summation, leading to the final expression of influence kernel $k$.

To further enhance the model expressiveness, we use a fully-connected neural network to represent each basis function. The history or displacement is taken as the input and fed through multiple hidden layers equipped with Softplus non-linear activation function. To allow for inhibiting influence from past events (negative value of influence kernel $k$), we use a linear output layer for each neural network. For an influence kernel with temporal rank $L$ and spatial rank $R$, we need $2(L + R)$ independent neural networks for modeling.

The benefits of our proposed kernel framework lies in the following: (i) The kernel parameterization with displacement significantly reduces the rank needed when representing the complicated kernel encountered in practice as shown in Figure 1. (ii) The non-stationarity of original influence of historical events over spatio-temporal space can be conveniently captured by in-homogeneous $\{\psi_l\}_{l=1}^{L}$, $\{u_r\}_{r=1}^{R}$, making the model applicable in modeling general STPPs. (iii) The propagating patterns of influence are characterized by $\{\varphi_l\}_{l=1}^{L}$, $\{v_r\}_{r=1}^{R}$ which go beyond simple parametric forms. In particular, when the events' influence has finite range, *i.e.* there exist $\tau_{\max}$ and $a_{\max}$ such that the influence decays to zero if $|t - t'| > \tau_{\max}$ or $||s - s'|| > a_{\max}$, we can restrict the parameterization of $\{\varphi_l\}_{l=1}^{L}$ and $\{v_r\}_{r=1}^{R}$ on a local domain $[0, \tau_{\max}] \times B(0, a_{\max})$ instead of $[0, T] \times \mathcal{S}$, which further reduce the model complexity. Details of choosing kernel and neural network architectures are described in Appendix C.

*Remark* 1 (the class of influence kernel expressed). The proposed deep kernel representation covers a large class of non-stationary kernels generally used in STPPs. In particular, the proposed form of the kernel does not need to be positive semi-definite or even symmetric (Reinhart, 2018). The low-rank decomposed formulation equation 4 is of SVD-type (cf. Appendix A.1). While each $\varphi_l$ (and $v_r$) can be viewed as stationary (i.e., shift-invariant), the combination with left modes in the summation enables to model spatio-temporal non-stationarity. The technical assumptions A.1 and A.2 do not require more than the existence of a low-rank decomposition motivated by kernel SVD. As long as the $2(R + L)$ many functions $\psi_l$, $\varphi_l$, and $u_r$, $v_r$ are sufficiently regular, they can be approximated and learned by a neural network. The universal approximation power of neural networks enables our framework to express a broad range of general kernel functions, and the low-rank decomposed form reduces the modeling of a spatio-temporal kernel to finite many functions on time and space domains (the right modes are on truncated domains), respectively.

## 4 EFFICIENT COMPUTATION OF MODEL

We consider model optimization through Maximum likelihood estimation (MLE) (Reinhart, 2018). The resulting conditional intensity function could now be negative by allowing inhibiting historical

influence. A common approach to guarantee the non-negativity is to adopt a nonlinear positive activation function in the conditional intensity (Du et al., 2016; Zhu et al., 2022). However, the integral of such a nonlinear intensity over spatio-temporal space is computationally expensive. To tackle this, we first introduce a log-barrier to the MLE optimization problem to guarantee the non-negativity of conditional intensity function $\lambda$ and maintain its linearity. Then we provide a computationally efficient strategy that benefits from the linearity of the conditional intensity. The extension of the approach to point process data with marks is given in Appendix B.

## 4.1 MODEL OPTIMIZATION WITH LOG-BARRIER

We re-denote $\ell(\mathcal{H})$ in equation 2 by $\ell(\theta)$ in terms of model parameter $\theta$. The constrained MLE optimization problem for model parameter estimation can be formulated as:

$$\min_{\theta} -\ell(\theta), s.t. -\lambda(t, s) \leq 0, \ \forall t \in [0, T], \forall s \in \mathcal{S}.$$

Introduce a log-barrier method (Boyd et al., 2004) to ensure the non-negativity of $\lambda$, and penalize the values of $\lambda$ on a dense enough grid $\mathcal{U}_{\text{bar},t} \times \mathcal{U}_{\text{bar},s} \subset [0, T] \times \mathcal{S}$. The log-barrier is defined as

$$p(\theta, b) := -\frac{1}{|\mathcal{U}_{\text{bar},t} \times \mathcal{U}_{\text{bar},s}|} \sum_{c_t=1}^{|\mathcal{U}_{\text{bar},t}|} \sum_{c_s=1}^{|\mathcal{U}_{\text{bar},s}|} \log(\lambda(t_{c_t}, s_{c_s}) - b), \tag{5}$$

where $c_t, c_s$ indicate the index of the gird, and $b$ is a lower bound of conditional intensity function on the grid to guarantee the feasibility of logarithm operation. The MLE optimization problem can be written as

$$\min_{\theta} \mathcal{L}(\theta) := -\ell(\theta) + \frac{1}{w} p(\theta, b) = -\left( \sum_{i=1}^{n} \log \lambda(t_i, s_i) - \int_0^T \int_{\mathcal{S}} \lambda(t, s) ds dt \right)$$

$$- \frac{1}{w|\mathcal{U}_{\text{bar},t} \times \mathcal{U}_{\text{bar},s}|} \sum_{c_t=1}^{|\mathcal{U}_{\text{bar},t}|} \sum_{c_s=1}^{|\mathcal{U}_{\text{bar},s}|} \log(\lambda(t_{c_t}, s_{c_s}) - b), \tag{6}$$

where $w$ is a weight to control the trade-off between log-likelihood and log-barrier; $w$ and $b$ can be set accordingly during the learning procedure. Details can be found in Appendix A.2.

Note that previous works (Du et al., 2016; Mei and Eisner, 2017; Pan et al., 2021; Zuo et al., 2020; Zhu et al., 2022) use a scaled positive transformation to guarantee non-negativity conditional intensity function. Compared with them, the log-barrier method preserves the linearity of the conditional intensity function. As shown in Table 1, such a log-barrier method enables efficient model computation (See more details in Section 4.2) and enhance the model recovery power.

## 4.2 MODEL COMPUTATION

The log-likelihood computation of general STPPs (especially those with general influence function) is often difficult and requires numerical integral and thus time-consuming. Given a sequence of events $\{x_i = (t_i, s_i)\}_{i=1}^{n}$ of number $n$, the complexity of neural network evaluation is of $\mathcal{O}(n^2)$ for the term of log-summation and of $\mathcal{O}(Kn)$ $(K \gg n)$ when using numerical integration for the double integral term with $K$ sampled points in a multi-dimensional space. In the following, we circumvent the calculation difficulty by proposing an efficient computation for $\mathcal{L}(\theta)$ with complexity $\mathcal{O}(n)$ of neural network evaluation through a domain discretization strategy.

**Computation of log-summation.** The first log-summation term in equation 2 can be written as:

$$\sum_{i=1}^{n} \log \lambda(t_i, s_i) = \sum_{i=1}^{n} \log \left( \mu + \sum_{t_j < t_i} \sum_{r=1}^{R} \sum_{l=1}^{L} \alpha_{lr} \psi_l(t_j) \varphi_l(t_i - t_j) u_r(s_j) v_r(s_i - s_j) \right). \tag{7}$$

Note that each $\psi_l$ only needs to be evaluated at event time $\{t_i\}_{i=1}^{n}$ and each $u_r$ is evaluated at all the event location $\{s_i\}_{i=1}^{n}$. To avoid the redundant evaluations of $\varphi_l$ over every pair of $(t_i, t_j)$, we set up a uniform grid $\mathcal{U}_t$ over time horizon $[0, \tau_{\max}]$ and evaluate $\varphi_l$ on the grid. The value of $\varphi_l(t_j - t_i)$ can be obtained by linear interpolation with values on two adjacent grid points of $t_j - t_i$. By doing

so, we only need to evaluate $\varphi_l$ for $|\mathcal{U}_t|$ times on the grids. Note that $\varphi_l$ can be simply feed with 0 when $t_j - t_i > \tau_{\max}$ without any neural network evaluation.

Here we directly evaluate $v_r(s_i - s_j)$ since numerical interpolation is less accurate in location space. Note that one does not need to evaluate every pair of index $(i, j)$. Instead, we have $I := \{(i,j) \mid v_r(s_i - s_j) \text{ will be computed}\} = \{(i,j) \mid t_j < t_i \le t_j + \tau_{\max}\} \cap \{(i,j) \mid \|s_i - s_j\| \le a_{\max}\}$. We use 0 to other pairs of $(i, j)$.

**Computation of integral.** A benefit of our approach is that we avoid numerical integration for the conditional intensity function (needed to evaluate the likelihood function), since the design of the kernel allows us to decompose the desired integral to integrating basis functions. Specifically, we have

$$\int_0^T \int_{\mathcal{S}} \lambda(t,s) ds dt = \mu |\mathcal{S}| T + \sum_{i=1}^{n} \int_0^T \int_{\mathcal{S}} I(t_i < t) k(t_i, t, s_i, s) ds dt$$

$$= \mu |\mathcal{S}| T + \sum_{i=1}^{n} \sum_{r=1}^{R} u_r(s_i) \int_{\mathcal{S}} v_r(s - s_i) ds \sum_{l=1}^{L} \alpha_{rl} \psi_l(t_i) \int_0^{T-t_i} \varphi_l(t) dt. \quad (8)$$

To compute the integral of $\varphi_l$, we take the advantage of the pre-computed $\varphi_l$ on the grid $\mathcal{U}_t$. Let $F_l(t) := \int_0^t \varphi_l(\tau) d\tau$. Then $F_l(T - t_i)$ can be computed by the linear interpolation of values of $F_l$ at two adjacent grid points (in $\mathcal{U}_t$) of $T - t_i$. In particular, $F_l$ evaluated on $\mathcal{U}_t$ equals to the cumulative sum of $\varphi_l$ divided by the grid width.

The integral of $v_r$ can be estimated based on a grid $\mathcal{U}_s$ in $B(0, a_{\max}) \subset \mathbb{R}^{d_S}$ since it decays outside the ball. For each $s_i$, $\int_{\mathcal{S}} v_r(s - s_i) ds = \int_{B(0, a_{\max}) \cap \{\mathcal{S} - s_i\}} v_r(s) ds$, where $\mathcal{S} - s_i := \{s' \mid s' = s - s_i, s \in \mathcal{S}\}$. Thus the integral is well estimated with the evaluations of $v_r$ on grid set $\mathcal{U}_s \cap \mathcal{S} - s_i$. Note that in practice we only evaluate $v_r$ on $\mathcal{U}_s$ once and use subsets of the evaluations for different $s_i$. More details about grid-based computation can be found in Appendix A.3.

**Computation of log-barrier.** The barrier term $p(\theta, b)$ is calculated in a similar way as equation 7 by replacing $(t_i, s_i, \mu)$ with $(t_{c_t}, s_{c_s}, \mu - b)$, *i.e.* we use interpolation to calculate $\varphi_l(t_{c_t} - t_j)$ and evaluate $v_r$ on a subset of $\{(s_{c_s}, s_j)\}$, $c_s = 1, \ldots, |\mathcal{U}_{\text{bar},s}|$, $j = 1, \ldots, n$.

### 4.3 COMPUTATIONAL COMPLEXITY

The evaluation of $\{u_r\}_{r=1}^{R}$ and $\{\psi_l\}_{l=1}^{L}$ over $n$ events costs $\mathcal{O}((R + L)n)$ complexity. The evaluation of $\{\varphi_l\}_{l=1}^{L}$ is of $\mathcal{O}(L|\mathcal{U}_t|)$ complexity since it relies on the grid $\mathcal{U}_t$. The evaluation of $\{v_r\}_{r=1}^{R}$ costs no more than $\mathcal{O}(RC\tau_{\max}n) + \mathcal{O}(R|\mathcal{U}_s|)$ complexity. We note that $L, R, \tau_{\max}, |\mathcal{U}_t|, |\mathcal{U}_s|$ are all constant that much less than event number $n$, thus

Table 1: Comparison of model training time per epoch on a 1D and a 3D synthetic data. Time is measured in second.

| Model | 1D Data set 1 | | 3D Data set 1 | |
|---|---|---|---|---|
| | #Parameters | Training time | #Parameters | Training time |
| NSMPP | 2576 | 60.040 | 9415 | 170.004 |
| DNSK+Barrier | 2307 | 1.299 | 9228 | 3.529 |

the overall computation complexity will be $\mathcal{O}(n)$. We compare the model training time per epoch for a baseline equipped with a softplus activation function (NSMPP) and our model with log-barrier method (DNSK+Barrier) on a 1D synthetic data set and a 3D synthetic data set. The quantitative results in Table 1 demonstrates the efficiency improvement of our model by using log-barrier technique. More details about the computation complexity analysis can be found in Appendix A.4.

## 5 EXPERIMENT

We use large-scale synthetic and real data sets to demonstrate the superior performance of our model and present the results in this section. Experimental details and results can be found in Appendix C. Codes will be released upon publication.

**Baselines.** We compare our method (DNSK+Barrier) with: (i) RMTPP (RMTPP) (Du et al., 2016); (ii) Neural Hawkes (NH) (Mei and Eisner, 2017); (iii) Transformer Hawkes process (THP) (Zuo et al., 2020); (iv) Parametric Hawkes process (PHP+exp) with exponentially decaying spatio-temporal kernel; (v) Neual spectral marked point processes (NSMPP) (Zhu et al., 2022); (vi) DNSK without log-barrier but with a non-negative Softplus activation function (DNSK+Softplus). We note that RMTPP, NH and THP directly model conditional intensity function using neural networks while others learn the influence kernel in the framework of equation 3. In particular, NSMPP designs

Table 2: Synthetic data results. Testing log-likelihood per event (on the left side of slash, higher the better) and MRE (on the right side of slash, lower the better) are reported in each entry.

| Model | 1D Data set 1 | 1D Data set 2 | 1D Data set 3 | 2D Data set 1 | 3D Data set 1 | 3D Data set 2 |
|---|---|---|---|---|---|---|
| RMTPP | $-0.467_{(0.009)}/0.086$ | $-2.591_{(0.010)}/0.259$ | $-1.353_{(0.002)}/0.212$ | $-5.268_{(0.053)}/0.195$ | $-2.561_{(0.015)}/0.117$ | $-2.289_{(0.002)}/0.316$ |
| NH | $-0.459_{(0.002)}/0.068$ | $-2.543_{(0.007)}/0.092$ | $-1.315_{(0.032)}/0.204$ | $-5.223_{(0.051)}/0.174$ | $-2.524_{(0.002)}/0.098$ | $-2.291_{(0.003)}/0.319$ |
| THP | $-0.537_{(0.008)}/0.843$ | $-2.554_{(0.005)}/0.106$ | $-1.319_{(0.003)}/0.115$ | $-5.292_{(0.029)}/0.182$ | $-2.527_{(0.001)}/0.041$ | $-2.497_{(0.018)}/0.350$ |
| PHP+exp | $-0.451_{(0.001)}/0.093$ | $-2.725_{(0.002)}/0.181$ | $-1.524_{(0.015)}/0.223$ | $-2.737_{(0.003)}/0.306$ | $-2.683_{(0.002)}/0.291$ | $-2.424_{(0.003)}/0.282$ |
| NSMPP | $-0.462_{(0.010)}/0.078$ | $-2.638_{(0.008)}/0.162$ | $-1.473_{(0.033)}/0.164$ | $-2.807_{(0.016)}/0.156$ | $-2.637_{(0.012)}/0.193$ | $-2.381_{(0.012)}/0.280$ |
| DNSK+Softplus | $-0.455_{(0.003)}/0.045$ | $-2.539_{(0.002)}/0.037$ | $-1.300_{(0.004)}/0.104$ | $-2.592_{(0.002)}/0.042$ | $-2.515_{(0.002)}/0.088$ | $-2.279_{(0.003)}/0.119$ |
| DNSK+Barrier | $\mathbf{-0.451_{(0.002)}/0.039}$ | $\mathbf{-2.536_{(0.003)}/0.016}$ | $\mathbf{-1.298_{(0.002)}/0.031}$ | $\mathbf{-2.585_{(0.001)}/0.028}$ | $\mathbf{-2.498_{(0.003)}/0.021}$ | $\mathbf{-2.251_{(0.001)}/0.082}$ |

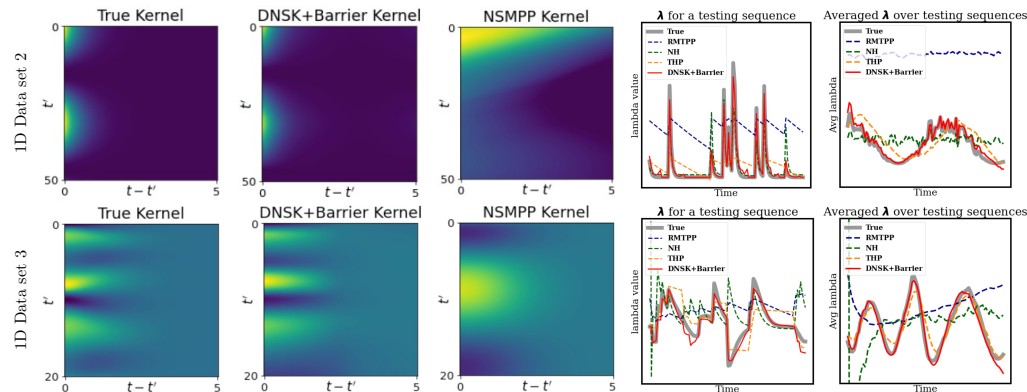

Figure 2: Kernel recovery results of 1D Data set 2 and 3. The first three columns show true kernels, kernels learned by `DNSK+Barrier`, and kernels learned by `NSMPP`, respectively. The last two columns show the predicted conditional intensity for a testing sequence and the averaged conditional intensity for all testing sequences in each data set, which can be regarded as the event intensity by taking the expectation of the history.

the kernel based on singular value decomposition but parameterizes it without displacement. The model parameters are estimated using the training data via Adam optimization method (Kingma and Ba, 2014). Details of training can be found in Appendix A.2 and C.

## 5.1 SYNTHETIC DATA EXPERIMENTS

**Synthetic data sets.** To show the effectiveness of `DNSK+Barrier`, we conduct all the models on three temporal data sets and three spatio-temporal data sets generated by following true kernels: (i) 1D exponential kernel (ii) 1D non-stationary kernel; (iii) 1D infinite rank kernel; (iv) 2D exponential kernel; (v) 3D non-stationary inhibition kernel; (vi) 3D non-stationary mixture kernel. Data sets are generated using thinning algorithm in Daley and Vere-Jones (2008). Each data set is composed of 2000 sequences. Details of kernel formulas and data generation can be found in Appendix C.

We consider two performance metrics for testing data evaluation: Mean relative error (MRE) of the predicted intensity and log-likelihood. The true and predicted $\lambda^*(x), \hat{\lambda}(x)$ can be calculated using equation 4 with true and learned kernel. The MRE for one test trajectory is defined as $\int_{\mathcal{X}} |\lambda^*(x) - \hat{\lambda}(x)|/\lambda^*(x)dx$ and the averaged MRE over all test trajectories is reported. The log-likelihood for observing each testing sequence can be computed according to equation 2, and the average predictive log-likelihood per event is reported. The log-likelihood shows the model's goodness-of-fit, and the intensity evaluation further reflects the model's ability to recover the underlying mechanism of event occurrence and predict the future.

The heat maps in Figure 2 visualize the results of non-stationary kernel recovery for `DNSK+Barrier` and `NSMPP` on 1D Data set 2 and 3 (The true kernel used in 1D Data set 3 is the one in Figure 1). `DNSK+Barrier` recovers the true kernel more accurately than `NSMPP`, indicating the strong representation power of the low-rank kernel parameterization with displacements. Line charts in Figure 2 present the recovered intensities with the true ones (dark grey curves). It demonstrates that our method can accurately capture the temporal dynamics of events. In particular, the average conditional intensity $\lambda$ over multiple testing sequences shows the model's ability to recover data non-stationarity over time. While `DNSK+Barrier` successfully captures the non-stationarity among data, both `RMTPP` and `NH` fail to do so by showing a flat curve of the averaged intensity. Note that `THP` with

Table 3: Results of real data sets with time and categorical marks. We compare the testing log-likelihood (Testing $\ell$, higher the better), event time prediction error (lower the better), and event type prediction accuracy (higher the better) of `DNSK+Barrier` and other baselines on each data set.

| Model | Financial | | | StackOverflow | | |
|---|---|---|---|---|---|---|
| | Testing $-\ell$ | Time RMSE | Type Accuracy | Testing $-\ell$ | Time RMSE | Type Accuracy |
| RMTPP | $-3.890$ | 1.560 | 0.620 | $-2.600$ | 9.780 | 0.459 |
| NH | $-3.600$ | 1.560 | 0.622 | $-2.550$ | 9.830 | 0.463 |
| THP | $-0.938$ | 1.019 | 0.596 | $\mathbf{-1.231}$ | 11.804 | 0.436 |
| NSMPP | $-3.058$ | 1.276 | 0.608 | $-3.182$ | 8.735 | 0.447 |
| DNSK+Softplus | $-0.889$ | 0.327 | 0.621 | $-2.173$ | 6.416 | 0.497 |
| DNSK+Barrier | $\mathbf{-0.709}$ | $\mathbf{0.153}$ | $\mathbf{0.630}$ | $-2.089$ | $\mathbf{4.833}$ | $\mathbf{0.508}$ |

positional encoding recovers the data non-stationarity (as shown in two figures in the last column). However, our method still outperforms `THP` which suffers from limited model expressiveness when complicated propagation of event influence is involved (see two figures in the penultimate column).

Tabel 2 summarized the quantitative results of testing log-likelihood and MRE. It shows that `DNSK+Barrier` has superior predictive performance against baselines in characterizing the dynamics of data generation in spatio-temporal space. Specifically, with evidently over-parameterization for 1D Data set 1 generated by a stationary exponentially decaying kernel, our model can still approximate the kernel and recover the true conditional intensity without overfitting, which shows the adaptiveness of our model. Moreover, `DNSK+Barrier` enjoys outstanding performance gain when learning a diverse variety of complicated non-stationary kernels. The comparison between `DNSK+Softplus` and `DNSK+Barrier` proves that the model with log-barrier achieves a better recovery performance by maintaining the linearity of the conditional intensity. `THP` outperforms `RMTPP` in non-stationary cases but is still limited due to its pre-assumed parametric form of influence propagation. More results about kernel and intensity recovery can be found in Appendix C.

## 5.2 REAL DATA RESULTS

**Real data sets.** We provide a comprehensive evaluation of our approach on several real-world data sets: we first use two popular data sets containing time-stamped events with categorical marks to demonstrate the robustness of `DNSK+Barrier` in marked STPPs (refer to Appendix B for detailed definition and kernel modeling): (i) *Financial Transactions.* (Du et al., 2016). This data set contains transaction records of a stock in one day with time unit milliseconds and the action (mark) of each transaction. We partition the events into different sequences by time stamps. (ii) *StackOverflow* (Leskovec and Krevl, 2014): The data is collected from the website StackOverflow over two years, containing reward records for users who promote engagement in the community. Each user's reward history is treated as a sequence.

Next, we demonstrate the practical versatility of the model using the following spatio-temporal data sets: (i) *Southern California earthquake data* provided by Southern California Earthquake Data Center (SCEDC) contains time and location information of earthquakes in Southern California. We collect 19,414 records from 1999 to 2019 with magnitude larger than 2.5 and partition the data into multiple sequences by month with average length of 40.2. (ii) *Atlanta robbery & burglary data*. Atlanta Police Department (APD) provides a proprietary data source for city crime. We extract 3420 reported robberies and 14958 burglaries with time and location from 2013 to 2019. Two crime types are preprocessed as separate data sets on a 10-day basis with average lengths of 13.7 and 58.7.

Finally, the model's ability to tackle high-dimensional marks is evaluated with *Atlanta textual crime data.* The proprietary data set provided by APD records 4644 crime incidents from 2016 to 2017 with time, location, and comprehensive text descriptions. The text information is preprocessed by TF-IDF technique, leading to a 5012-dimensional mark for each event.

Table 3 summarizes the results of models dealing with categorical marks. Event time and type prediction are evaluated by Root Mean Square Error (RMSE) and accuracy, respectively. We can see that `DNSK+Barrier` outperforms the baselines in all prediction tasks by providing less time RMSE and higher type accuracy.

For real-world spatio-temporal data, we report average predictive log-likelihood per event for the testing set since MRE is not applicable. Besides, we perform *online prediction* for earthquake data to

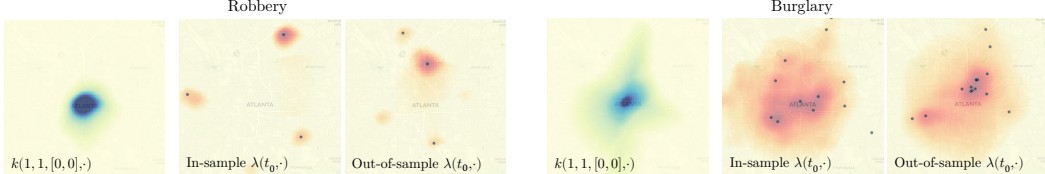

Figure 3: Kernel fitting and intensity prediction for ATL robbery & burglary data. First panel in each category shows the learned influence kernel of each crime at fixed geolocation in downtown ATL. Other panels show the predicted conditional intensity for two crimes over space at an in-sample and out-of-sample time, respectively. The dots represent reported events on that day.

Table 4: Real data results with spatial and mark information. Testing log-likelihood (higher the better) and prediction mean absolute error of event time and location (lower the better) are reported.

| | South California Earthquake | | | Atlanta Robbery | Atlanta Burglary | Atlanta Textual Crime |
|---|---|---|---|---|---|---|
| Model | Testing $\ell$ | Time MAE | Location MAE | Testing $\ell$ | Testing $\ell$ | Testing $\ell$ |
| RMTPP | $-1.825_{(0.053)}$ | $6.963$ | $0.602$ | $-2.349_{(0.018)}$ | $-0.246_{(0.008)}$ | / |
| NH | $-1.818_{(0.037)}$ | $6.880$ | $0.458$ | $-2.445_{(0.016)}$ | $-0.308_{(0.012)}$ | / |
| THP | $-1.784_{(0.007)}$ | $6.113$ | $0.633$ | $\mathbf{-2.221}_{(0.005)}$ | $-0.251_{(0.001)}$ | / |
| PHP+exp | $-2.048_{(0.093)}$ | $8.132$ | $0.487$ | $-2.641_{(0.006)}$ | $0.382_{(0.005)}$ | $-10.542_{(0.016)}$ |
| NSMPP | $-4.152_{(0.187)}$ | $6.780$ | $0.455$ | $-2.753_{(0.113)}$ | $0.304_{(0.098)}$ | $-8.694_{(0.336)}$ |
| DNSK+Softplus | $-1.807_{(0.096)}$ | $1.685$ | $0.492$ | $-2.435_{(0.009)}$ | $0.493_{(0.008)}$ | $-5.876_{(0.057)}$ |
| DNSK+Barrier | $\mathbf{-1.751}_{(0.080)}$ | $\mathbf{1.474}$ | $\mathbf{0.431}$ | $-2.255_{(0.008)}$ | $\mathbf{0.519}_{(0.012)}$ | $\mathbf{-5.279}_{(0.044)}$ |

[1] RMTPP, NH, and THP are not applicable when dealing with high-dimensional data.

demonstrate the model predicting ability. The probability density function $f(t, s)$ which represents the conditional probability that the next event will occur at $(t, s)$ given history $\mathcal{H}_t$ can be written as $f(t, s) = \lambda(t, s) \exp \left( - \int_{\mathcal{S}} \int_{t_n}^{t} \lambda(\tau, \nu) d\tau d\nu \right)$. The predicted time and location of the next event can be computed as $\mathbb{E}\left[ t_{n+1} | \mathcal{H}_t \right] = \int_{t_n}^{\infty} t \int_{\mathcal{S}} f(t, s) ds dt$, $\mathbb{E}\left[ s_{n+1} | \mathcal{H}_t \right] = \int_{\mathcal{S}} s \int_{t_n}^{\infty} f(t, s) dt ds$. We predict the the time and location of the last event in each sequence. The mean absolute error (MAE) of the predictions is computed. The quantitative results in Table 4 show that DNSK+Barrier provides more accurate predictions than other alternatives with higher event log-likelihood.

To demonstrate our model's interpretability and power to capture heterogeneous data characteristics, we visualize the learned influence kernels and predicted conditional intensity for two crime categories in Figure 3. The first column shows kernel evaluations at fixed geolocation in downtown Atlanta which intuitively reflect the spatial influence of crimes in that neighborhood. The influence of a robbery in the downtown area is more intensive but regional, while a burglary which is hard to be responded to by police in time would impact a larger neighborhood along major highways of Atlanta. We also provide the predicted conditional intensity over space for two crimes. As we can observe, DNSK+Barrier captures the occurrence of events in regions with a higher crime rate, and crimes of the same category happening in different regions would influence their neighborhoods differently. We note that this example emphasizes the ability of the proposed method to recover data non-stationarity with different sequence lengths, and improve the limited model interpretability of other neural network-based methods (RMTPP, NH, and THP) in practice.

For Atlanta textual crime data, we borrow the idea in Zhu and Xie (2022) by encoding the highly sparse TF-IDF representation into a binary mark vector with dimension $d = 50$ using Restricted Boltzmann Machine (RBM) (Fischer and Igel, 2012). The average testing log-likelihoods per event for each model are reported in Table 4. The results show that DNSK+Barrier outperforms PHP+exp in Zhu and Xie (2022) and NSMPP by achieving a higher testing log-likelihood. We visualize the basis functions of learned influence kernel by DNSK+Barrier in Figure A.4 in Appendix.

## 6 CONCLUSION

We propose a deep non-stationary kernel for spatio-temporal point processes using a low-rank parameterization based on displacement, which enables the model to be further low-rank when learning complicated influence kernel and significantly reduces model complexity. The non-negativity of the intensity is guaranteed by a log-barrier method that maintains the linearity of the conditional intensity function. Based on that, we propose a computationally efficient strategy for model estimation. The superior performance of our model is demonstrated using synthetic and real data sets.

ACKNOWLEDGEMENT

The work is partially supported by NSF DMS-2134037. Z.D. and Y.X. are partially supported by an NSF CAREER CCF-1650913, and NSF DMS-2134037, CMMI-2015787, CMMI-2112533, DMS-1938106, and DMS-1830210. X.C. is partially supported by NSF and the Alfred P. Sloan Foundation.

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

# A  ADDITIONAL METHODOLOGY DETAILS

## A.1  DERIVATION OF EQUATION 4

We denote $\tau := t-t'$, $\nu := s-s'$, the variables $t' \in [0, T]$, $\tau \in [0, \tau_{\max}]$, $s' \in \mathcal{S}$ and $\nu \in B(0, a_{\max})$, where the sets $\mathcal{S}, B(0, a_{\max}) \subset \mathbb{R}^2$. Viewing the spatial and temporal variables, i.e., $(t', \tau)$ and $(s', \nu)$, as left and right mode variables, respectively, the kernel function SVD (Mollenhauer et al., 2020; Mercer, 1909) of $k$ gives that

$$k(t', \tau, s', \nu) = \sum_{k=1}^{\infty} \sigma_k g_k(t', \tau) h_k(s', \nu). \tag{A.1}$$

We assume that the SVD can be truncated at $k \leq K$ with a residual of $\varepsilon$ for some small $\varepsilon > 0$, and this holds as long as the singular values $\sigma_k$ decay sufficiently fast. To fulfill the approximate finite-rank representation, it suffices to have the scalars $\sigma_k$ and the functions $g_k$ and $h_k$ so that the expansion approximates the kernel $k$, even if they are not SVD of the kernel. This leads to the following assumption:

**Assumption A.1.** *There exist coefficients $\sigma_k$, and functions $g_k(t', \tau)$, $h_k(s', \nu)$ s.t.*

$$k(t', \tau, s', \nu) = \sum_{k=1}^{K} \sigma_k g_k(t', \tau) h_k(s', \nu) + O(\varepsilon). \tag{A.2}$$

To proceed, one can apply kernel SVD again to $g_k$ and $h_k$ respectively, and obtain left and right singular functions that potentially differ for different $k$. Here, we impose that *across $k = 1, \cdots, K$, the singular functions of $g_k$ are the same* (as shown below, being approximately same suffices) set of basis functions, that is,

$$g_k(t', \tau) = \sum_{l=1}^{\infty} \beta_{k,l} \psi_l(t') \varphi_l(\tau).$$

As we will truncate $l$ to be up to a finite rank again (up to an $O(\varepsilon)$ residual) we require the (approximately) shared singular modes only up to $L$. Similarly as above, technically it suffices to have a finite-rank expansion to achieve the $O(\varepsilon)$ error without requiring them to be SVD, which leads to the following assumption where we assume the same condition for $h_k$:

**Assumption A.2.** *For the $g_k$ and $h_k$ in equation A.2, up to an $O(\varepsilon)$ error,*

*(i) The $K$ temporal kernel functions $g_k(t', \tau)$ can be approximated under a same set of left and right basis functions, i.e., there exist coefficients $\beta_{kl}$, and functions $\psi_l(t')$, $\varphi_l(\tau)$ for $l = 1, \cdots, L$, s.t.*

$$g_k(t', \tau) = \sum_{l=1}^{L} \beta_{kl} \psi_l(t') \varphi_l(\tau) + O(\varepsilon), \quad k = 1, \cdots, K. \tag{A.3}$$

*(ii) The $K$ spatial kernel functions $h_k(s', \nu)$ can be approximated under a same set of left and right basis functions, i.e., there exist coefficients $\gamma_{kr}$, and functions $u_r(s')$, $v_r(\nu)$ for $r = 1, \cdots, R$, s.t.*

$$h_k(s', \nu) = \sum_{r=1}^{R} \gamma_{kr} u_r(t') v_r(\nu) + O(\varepsilon), \quad r = 1, \cdots, R. \tag{A.4}$$

Inserting equation A.3 and equation A.4 into equation A.2 gives the rank-truncated representation of the kernel function. Since $K, L, R$ are fixed numbers, assuming boundedness of all the coefficients and functions, we have the representation with the final residual as $O(\varepsilon)$, namely,

$$k(t', \tau, s', \nu) = \sum_{l=1}^{L} \sum_{r=1}^{R} \sum_{k=1}^{K} \sigma_k \beta_{kl} \gamma_{kr} \psi_l(t') \varphi_l(\tau) u_r(t') v_r(\nu) + O(\varepsilon).$$

Defining $\sum_{k=1}^{K} \sigma_k \beta_{kl} \gamma_{kr}$ as $\alpha_{lr}$ leads to equation 4.

## A.2  ALGORITHMS

---

**Algorithm 1** Model parameter estimation

---

**Input**: Training set $X$, batch size $M$, epoch number $E$, learning rate $\gamma$, constant $a > 1$ to update $s$ in equation 6.
**Initialization:** model parameter $\theta_0$, first epoch $e = 0$, $s = s_0$.
**while** $e < E$ **do**
    **for** each batch with size $M$ **do**
        1. For 1D temporal point process, compute $\ell(\theta), \{\lambda(t_{c_t})\}_{c_t=1,...,|\mathcal{U}_{\mathrm{bar},t}|}$. For spatio-temporal point process, compute $\ell(\theta), \{\lambda(t_{c_t}, s_{c_s})\}_{c_t=1,...,|\mathcal{U}_{\mathrm{bar},t}|, c_s=1,...,|\mathcal{U}_{\mathrm{bar},s}|}$.

        2. Set $b = \min\{\lambda(t_{c_t})\}_{c_t=1,...,|\mathcal{U}_{\mathrm{bar},t}|} - \epsilon$ (or $\min\{\{\lambda(t_{c_t}, s_{c_s})\}_{c_t=1,...,|\mathcal{U}_{\mathrm{bar},t}|, c_s=1,...,|\mathcal{U}_{\mathrm{bar},s}|} - \epsilon$), where $\epsilon$ is a small value to guarantee logarithm feasibility.

        3. Compute $\mathcal{L}(\theta) = -\ell(\theta) + \frac{1}{w}p(\theta, b)$.

        4. Update $\theta_{e+1} \leftarrow \theta_e - \gamma \frac{\partial \mathcal{L}}{\partial \theta_e}$.

        5. $e \leftarrow e + 1, w \leftarrow w \cdot a$
    **end for**
**end while**

---

**Algorithm 2** Synthetic data generation

---

**Input:** Model $\lambda(\cdot), T, \mathcal{S}$, Upper bound of conditional intensity $\bar{\lambda}$.
**Initialization:** $\mathcal{H}_T = \emptyset, t = 0, n = 0$
**while** $t < T$ **do**
    1. Sample $u \sim \mathrm{Unif}(0, 1)$.
    2. $t \leftarrow t - \ln u / \bar{\lambda}$.
    3. Sample $s \sim \mathrm{Unif}(\mathcal{S}), D \sim \mathrm{Unif}(0, 1)$.
    4. $\lambda = \lambda(t, s | \mathcal{H}_T)$.
    **if** $D\bar{\lambda} \leq \lambda$ **then**
        $n \leftarrow n + 1; t_n = t, s_n = s$.
        $\mathcal{H}_T \leftarrow \mathcal{H}_T \cup \{(t_n, s_n)\}$.
    **end if**
**end while**
**if** $t_n >= T$ **then**
    **return** $\mathcal{H}_T - \{(t_n, s_n)\}$
**else**
    **return** $\mathcal{H}_T$
**end if**

---

### A.3 GRID-BASED MODEL COMPUTATION

In this section, we elaborate on the details of the grid-based efficient model computation.

In Figure A.1, we visualize the procedure of computing the integrals of $\int_0^{T-t_i} \varphi_l(t)dt$ and $\int_{\mathcal{S}} v_r(s - s_i)ds$ in equation 8, respectively. Panel (a) illustrates the calculation of $\int_0^{T-t_i} \varphi_l(t)dt$. As explained in Section 4.2, the evaluations of $\varphi_l$ only happens on the grid $\mathcal{U}_t$ over $[0, \tau_{\max}]$ (since $\varphi_l(t) = 0$ when $t > \tau_{\max}$). The value of $F(t) = \int_0^t \varphi_l(\tau)d\tau$ on the grid can be obtained through numerical integration. Then given $t_i$, the value of $F(T - t_i) = \int_0^{T-t_i} \varphi_l(t)dt$ is calculated using linear interpolation of $F$ on two adjacent grid points of $T - t_i$. Panel (b) shows the computation of $\int_{\mathcal{S}} v_r(s - s_i)ds$. Given $s_i, \int_{\mathcal{S}} v_r(s - s_i)ds = \int_{B(0,a_{\max}) \cap \{\mathcal{S}-s_i\}} v_r(s)ds$ since $v_r(s) = 0$ when $s > a_{\max}$. Then $B(0, a_{\max})$ is discretized into the grid $\mathcal{U}_s$, and $\int_{\mathcal{S}} v_r(s - s_i)ds$ can be calculated based on the value of $v_r$ on the grid points in $\mathcal{U}_s \cap \mathcal{S} - s_i$ (the deep red dots in Figure A.1(b)) using numerical integration.

To evaluate the sensitivity of our model to the chosen grids, we compare the performance of `DNSK+Barrier` on 3D Data set 2 using grids with different resolutions. The quantitative results of testing log-likelihood and intensity prediction error are reported in Table A.1. We use $|\mathcal{U}_t| = 50, |\mathcal{U}_s| = 1500$ for the experiments in the main paper. As we can see, the model shows similar performances when a higher grid resolution is used and works slightly less accurately but still

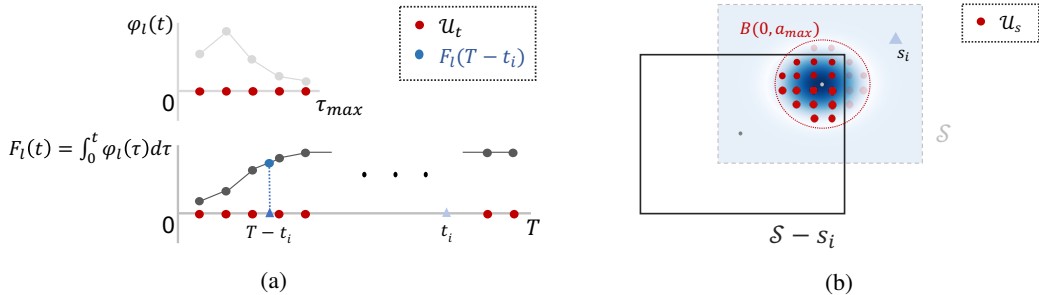

(a)                                                                (b)

Figure A.1: (a) Computation of $\int_0^{T-t_i} \varphi_l(t)dt$ computation based on grid $\mathcal{U}_t$. Red dots represent grid points. Line segments connecting two light or dark grey dots represent the linear interpolation of $\varphi_l$ and $F_l$. Here $t_i$ is the time of the historical event which is fixed. (b) Computation of $\int_{\mathcal{S}} v_r(s - s_i)ds$ based on grid $\mathcal{U}_s$. The background heatmap represents the evaluation of $v_r$ over $\mathcal{S}$. Here the fixed $s_i$ is the location of the historical event. The integral is calculated based on the values of $v_r$ on grid points with dark red color.

Table A.1: Comparison of `DNSK+Barrier` performance on 3D Data set 2 with different grid resolutions. Testing log-likelihood per event and intensity MRE are reported. The highlighted ones are the results in the main paper.

| Temporal resolution: $|\mathcal{U}_t|$ | Spatial resolution: $|\mathcal{U}_s|$ | | |
|---|---|---|---|
| | 1000 | 1500 | 3000 |
| 30 | $-2.272_{(0.005)}/0.102$ | $-2.252_{(0.002)}/0.088$ | $-2.250_{(0.002)}/0.081$ |
| 50 | $-2.257_{(0.002)}/0.095$ | $-2.251_{(0.001)}/0.082$ | $-2.249_{(0.001)}/0.078$ |
| 100 | $-2.255_{(0.001)}/0.091$ | $-2.252_{(0.001)}/0.081$ | $-2.250_{(0.001)}/0.078$ |

better than other baselines with less number of grid points. It reveals that our choice of grid resolution is accurate enough to capture the complex dynamics of event occurrences for this non-stationary data, and the model performance is robust to different grid resolutions.

In practice, the grids can be flexibly chosen to reach the balance of model accuracy and computational efficiency. For instance, the number of uniformly distributed grid points along one dimension can be chosen around $\mathcal{O}(n_0)$, where $n_0$ is the average number of events in one observed sequence. Note that $|\mathcal{U}_t|$ or $|\mathcal{U}_s|$ would be far less than the total number of observed events because we use thousands of sequences (2000 in our synthetic experiments) for model learning. And the grid size can be even smaller when it comes to non-Lebesgue-measured space.

## A.4    DETAILS OF COMPUTATIONAL COMPLEXITY

We provide the detailed analysis of the $\mathcal{O}(n)$ computation complexity of $\mathcal{L}(\theta)$ in Section 4.3 as following:

• Computation of log-summation. The evaluation of $\{u_r\}_{r=1}^R$ and $\{\psi_l\}_{l=1}^L$ over $n$ events costs $\mathcal{O}((R+L)n)$ complexity. The evaluation of $\{\varphi_l\}_{l=1}^L$ is of $\mathcal{O}(L|\mathcal{U}_t|)$ complexity since it relies on the grid $\mathcal{U}_t$. With the assumption that the conditional intensity is bounded by a constant $C$ in a finite time horizon (Lewis and Shedler, 1979; Daley et al., 2003; Zhu et al., 2022), for each fixed $j$, the cardinality of set $\{(i, j) \mid t_j < t_i \leq t_j + \tau_{\max}\}$ is less than $C\tau_{\max}$, which leads to a $\mathcal{O}(RC\tau_{\max}n)$ complexity of $\{v_r\}_{r=1}^R$ evaluation.

• Computation of integral. The integration of $\{\varphi_l\}_{l=1}^L$ only relies on numerical operations of $\{\varphi_l\}_{l=1}^L$ on grids $\mathcal{U}_t$ without extra evaluations of neural networks. The integration of $\{v_r\}_{r=1}^R$ depends on the evaluation on grid $\mathcal{U}_s$ of $\mathcal{O}(R|\mathcal{U}_s|)$ complexity.

• Computation of barrier. $\{\varphi_l\}_{l=1}^L$ on grid $\mathcal{U}_{\text{bar},t}$ is estimated by numerical interpolation of previously computed $\{\varphi_l\}_{l=1}^L$ on grid $\mathcal{U}_t$. Additional neural network evaluations of $\{v_r\}_{r=1}^R$ cost no more than $\mathcal{O}(RC\tau_{\max}n)$ complexity.

# B   DEEP NON-STATIONARY KERNEL FOR MARKED STPPS

In marked STPPs (Reinhart, 2018), each observed event is associated with additional information describing event attribute, denoted as $m \in \mathcal{M} \subset \mathbb{R}^{d_{\mathcal{M}}}$. Let $\mathcal{H} = \{(t_i, s_i, m_i)\}_{i=1}^n$ denote the event sequence. Given the observed history $\mathcal{H}_t = \{(t_i, s_i, m_i) \in \mathcal{H} | t_i < t\}$, the conditional intensity function of a marked STPPs is similarly defined as:

$$\lambda(t, s, m) = \lim_{\Delta t \downarrow 0, \Delta s \downarrow 0, \Delta m \downarrow 0} \frac{\mathbb{E}\left[\mathbb{N}([t, t + \Delta t] \times B(s, \Delta s) \times B(m, \Delta m)) \mid \mathcal{H}_t\right]}{|B(s, \Delta s)||B(m, \Delta m)|\Delta t},$$

where $B(m, \Delta m)$ is a ball centered at $m \in \mathbb{R}^{d_{\mathcal{M}}}$ with radius $\Delta m$. The log-likelihood of observing $\mathcal{H}$ on $[0, T] \times \mathcal{S} \times \mathcal{M}$ is given by

$$\ell(\mathcal{H}) = \sum_{i=1}^n \log \lambda(t_i, s_i, m_i) - \int_0^T \int_{\mathcal{S}} \int_{\mathcal{M}} \lambda(t, s, m) dm ds dt.$$

## B.1   KERNEL INCORPORATING MARKS

One of the salient features of our spatio-temporal kernel framework is that it can be conveniently adopted in modeling marked STPPs with additional sets of mark basis functions $\{g_q, h_q\}_{q=1}^Q$. We modify the influence kernel function $k$ accordingly as following:

$$k(t', t - t', s', s - s', m', m) = \sum_{q=1}^Q \sum_{r=1}^R \sum_{l=1}^L \alpha_{lrq} \psi_l(t') \varphi_l(t - t') u_r(s') v_r(s - s') g_q(m') h_q(m).$$

Here $m', m \in \mathcal{M} \subset \mathbb{R}^{d_{\mathcal{M}}}$ and $\{g_q, h_q : \mathcal{M} \to \mathbb{R}, q = 1, \ldots, Q\}$ represented by independent neural networks model the influence of historical mark $m'$ and current mark $m$, respectively. Since the mark space $\mathcal{M}$ is always categorical and the difference between $m'$ and $m$ is of little practical meaning, we use $g_q$ and $h_q$ to model $m'$ and $m$ separately instead of modeling $m - m'$.

## B.2   LOG-BARRIER AND MODEL COMPUTATION

The conditional intensity for marked spatio-temporal point processes at $(t, s, m)$ can be written as:

$$\lambda(t, s, m) = \mu + \sum_{l,r,q} \alpha_{lrq} \sum_{(t_i, s_i, m_i) \in \mathcal{H}_t} \psi_l(t_i) \varphi(t - t_i) u_r(s_i) v_r(s - s_i) g_q(m_i) h_q(m).$$

We need to guarantee the non-negativity of $\lambda$ over the space of $[0, T] \times \mathcal{S} \times \mathcal{M}$. When the total number of unique categorical mark in $\mathcal{M}$ is small, the log-barrier can be conveniently computed as the summation of $\lambda$ on grids $\mathcal{U}_{\text{bar},t} \times \mathcal{U}_{\text{bar},s} \times \mathcal{M}$. In the following we focus on the case that $\mathcal{M}$ is high-dimensional with $\mathcal{O}(n)$ number of unique marks.

For model simplicity we use non-negative $g_q$ and $h_q$ in this case (which can be done by adding a non-negative activation function to the linear output layer in neural networks). We re-write $\lambda(t, s, m)$ and denote as following:

$$\lambda(t, s, m) = \mu + \sum_q \underbrace{\left( \sum_{l,r} \alpha_{lrq} \sum_{(t_i, s_i, m_i) \in \mathcal{H}_t} \psi_l(t_i) \varphi(t - t_i) u_r(s_i) v_r(s - s_i) g_q(m_i) \right)}_{\hat{F}_q(t,s)} h_q(m).$$

Note that the function in the brackets are only with regard to $t, s$. We denote it as $\hat{F}_q(t, s)$ (since it is in the $r$th rank of mark). Since $h_q(m) \geq 0$, the non-negativity of $\lambda$ can be guaranteed by the non-negativity of $\hat{F}_q(t, s)$. Thus we apply log-barrier method on $\hat{F}_q(t, s)$. The log-barrier term becomes:

$$p(\theta, b) := -\frac{1}{Q|\mathcal{U}_{\text{bar},t} \times \mathcal{U}_{\text{bar},s}|} \sum_{c_t=1}^{|\mathcal{U}_{\text{bar},t}|} \sum_{c_s=1}^{|\mathcal{U}_{\text{bar},s}|} \sum_{q=1}^Q \log(\hat{F}_q(t_{c_t}, s_{c_s}) - b),$$

Since our model is low-rank, the value of $Q$ will not be large.

For the model computation, the additional evaluations for $\{g_q\}_{q=1}^Q$ on events is of $\mathcal{O}(Qn)$ complexity and the evaluations for $\{h_q\}_{q=1}^Q$ only depends on the unique number of marks which at most of $\mathcal{O}(n)$. The log-barrier method does not introduce extra evaluation in mark space. Thus the overall computation complexity for `DNSK` in marked STPPs is still $\mathcal{O}(n)$.

## C ADDITIONAL EXPERIMENTAL RESULTS

In this section we provide details of data sets and experimental setup, together with additional experimental results.

**Synthetic data sets.** To show the robustness of our model, we generate three temporal data sets and three spatio-temporal data sets using the following kernels:

(i) 1D Data set 1 with exponential kernel: $k(t', t) = 0.8e^{-(t-t')}$.

(ii) 1D Data set 2 with non-stationary kernel: $k(t', t) = 0.3(0.5 + 0.5\cos(0.2t'))e^{-2(t-t')}$.

(iii) 1D Data set 3 with infinite rank kernel:

$$k(t', t) = 0.3\sum_{j=1}^{\infty} 2^{-j}\left(0.3 + \cos(2 + (\frac{t'}{5})^{0.7}1.3(j+1)\pi)\right)e^{-\frac{8(t-t')^2}{25}j^2}$$

(iv) 2D Data set 1 with exponential kernel: $k(t', t, s', s) = 0.5e^{-1.5(t-t')}e^{-0.8s'}$.

(v) 3D Data set 1 with non-stationary inhibition kernel:

$$k(t', t, s', s) = 0.3(1 - 0.01t)e^{-2(t-t')}\frac{1}{2\pi\sigma_{s'}^2}e^{-\frac{\|s'\|^2}{2\sigma_{s'}^2}}\frac{\cos(10\|s - s'\|)}{2\pi\sigma_s^2(1 + e^{10(\|s-s'\|-0.5)})}e^{-\frac{\|s-s'\|^2}{2\sigma_s^2}}$$

, where $\sigma_{s'} = 0.5, \sigma_s = 0.15$.

(vi) 3D Data set 2 with non-stationary mixture kernel:

$$k(t', t, s', s) = \sum_{r=1}^{2}\sum_{l=1}^{2}\alpha_{rl}u_r(s')v_r(s - s')\psi_l(t')\varphi_l(t - t')$$

, where $u_1(s') = 1 - a_s(s_2' + 1), u_2(s') = 1 - b_s(s_2' + 1), v_1(s - s') = \frac{1}{2\pi\sigma_1^2}e^{-\frac{\|s-s'\|^2}{2\sigma_1^2}}, v_2(s - s') = \frac{1}{2\pi\sigma_2^2}e^{-\frac{\|s-s'-0.8\|^2}{2\sigma_2^2}}, \psi_1(t') = 1 - a_t t', \psi_2(t') = 1 - b_t t', \varphi_1(t - t') = e^{-\beta(t-t')}, \varphi_2(t - t') = (t - t' - 1) \cdot I(t - t' < 3)$, and $a_s = 0.3, b_s = 0.4, a_t = 0.02, b_t = 0.02, \sigma_1 = 0.2, \sigma_2 = 0.3, \beta = 2, (\alpha_{11}, \alpha_{12}, \alpha_{21}, \alpha_{22}) = (0.6, 0.15, 0.225, 0.525)$.

Note that kernel (iii) is the one we illustrated in Figure 1, which is of infinite rank according to the formulas. In Figure 1, the value matrix of $k(t', t)$ and $k(t', t - t')$ are the kernel evaluations on a same $300 \times 300$ uniform grid. As we can see, the rank of the value matrix of the same kernel $k$ is reduced from 298 to 7 after changing to the displacement-based kernel parameterization.

**Details of Experimental setup.** For `RMTPP` and `NH` we test embedding size of $\{32, 64, 128\}$ and choose 64 for experiments. For `THP` we take the default experiment setting recommended by Zuo et al. (2020). For `NSMPP` we use the same model setting in Zhu et al. (2022) with rank 5. Each experiment is implemented by the following procedure: Given the data set, we split $90\%$ of the sequences as training set and $10\%$ as testing set.

We use independent fully-connected neural networks with two-hidden layers for each basis function. Each layer contains 64 hidden nodes. The temporal rank of `DNSK+Barrier` is set to be 1 for synthetic data (i), (ii), (iv), (v), 2 for (vi), and 3 for (iii). The spatial rank is 1 for synthetic data (iv), (v) and 2 for (vi). The temporal and spatial rank for real data are both set to be 2 through cross

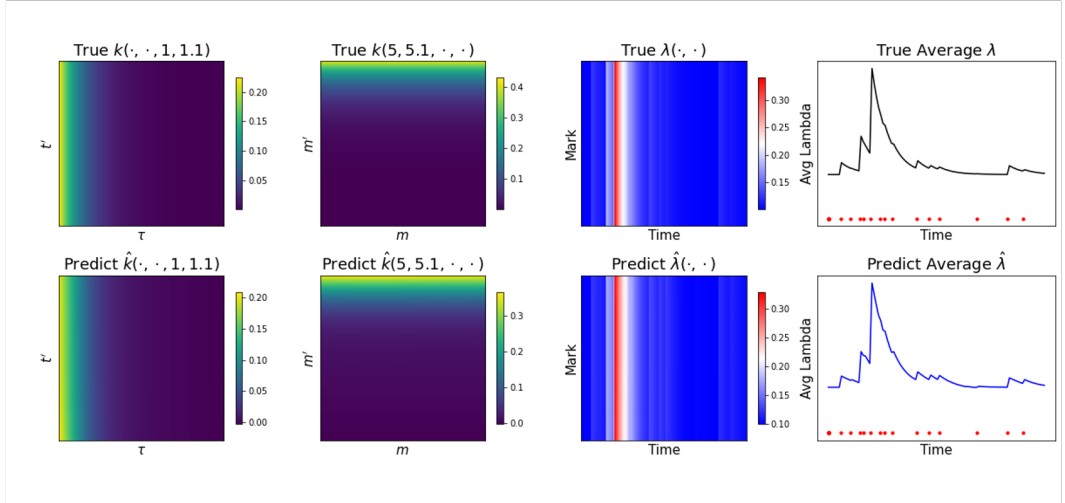

Figure A.2: Kernel recovery results of 2D exponential kernel. The first two columns show the true kernel and kernel learned by `DNSK+Barrier`. The last two columns shows the true and predicted conditional intensity functions of a test sequence. The line charts visualize the conditional intensity average over the 1D mark space at any given time for the ease of presentation. The red dots indicate the time of observed events.

validation. For each real data set, the $\tau_{\max}$ is chosen to be around $T/4$ and $s_{\max}$ is 1 for each data set since the location space is normalized before training. The hyper-parameter of `DNSK+Softplus` are the same as `DNSK+Barrier`. For `RMTPP`, `NH`, and `THP` the batch size is 32 and the learning rate is $10^{-3}$. For others, the batch size is 64 and the learning rate is $10^{-1}$. The quantitative results are collected by running each experiment for 5 independent times. All experiments are implemented on Google Colaboratory (Pro version) with 25GB RAM and a Tesla T4 GPU.

## C.1 SYNTHETIC RESULTS WITH 2D & 3D KERNEL

In this section we present additional experiment results for the synthetic data sets with 2D exponential and 3D non-stationary mixture kernel. Our proposed model successfully recovers the kernel and event conditional intensity in both case. Note that the recovery of 3D mixture kernel demonstrates the capability of our model to handle complex event dependency with mixture patterns by conveniently setting time and mark rank to be more than 1.

## C.2 ATLANTA TEXTUAL CRIME DATA WITH HIGH-DIMENSIONAL MARKS

Figure A.4 visualizes the fitting and prediction results of `DNSK+Barrier`. Our model presents an decaying pattern in temporal effect and captures two different patterns of spatial influence for incidents in the northeast. Besides, the in-sample and out-of-sample intensity predictions demonstrate the ability of `DNSK` to characterize the event occurrences by showing different conditional intensities.

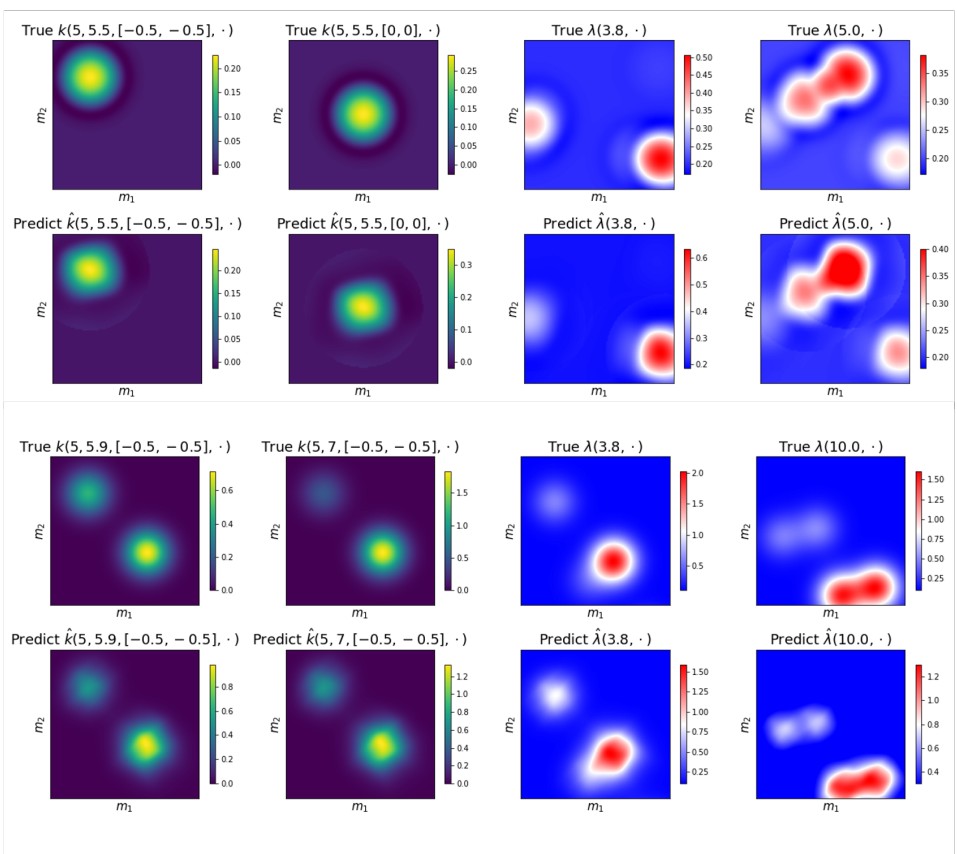

Figure A.3: Kernel recovery results of 3D non-stationary mixture kernel. The first two columns show snapshots of the true kernel and kernel learned by `DNSK+Barrier`. The last two columns shows snapshots of the true and predicted conditional intensity functions of a test sequence.

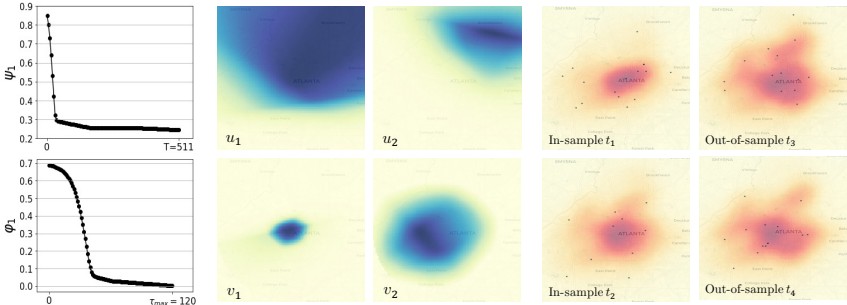

Figure A.4: Model fitting and prediction for high-dimensional real data. First column shows the learned temporal functions. Four panels in the middle shows the learned spatial functions, Deeper color depth indicates higher function value. Last four panels show the predicted conditional intensity over space at two in-sample times and two out-of-sample times, respectively. The dots represent event occurrences at that day.

