# OpenReview forum: "Spatio-temporal point processes with deep non-stationary kernels"
_ICLR.cc/2023/Conference — ICLR 2023 poster_

### Official Review · Reviewer_GLBW · 2022-10-23

**Confidence:** 4
**Correctness:** 3
**Technical Novelty And Significance:** 2
**Empirical Novelty And Significance:** 2
**Recommendation:** 8

**Clarity, Quality, Novelty And Reproducibility:**

DNSK appears as an extension of previous neural process models with "displacement"-based factorized neural kernels. Such an implementation does provide some performance improvement based on the reported experiments.

Overall presentation is reasonable clear. The authors may want to consider further improve the presentation, for example, better justify the adopted kernels. There are also several language problems throughout the submission: for example. $B(s,\Delta s)$ seems to be a ball set with the center $s$ and radius $\Delta s$ but is not defined; symbol $f$ is used as transformation in page 3 but as pdf of the event in page 8;  and many others throughout the submission.

**Strength And Weaknesses:**

Efficient modeling non-stationary spatio-temporal processes is challenging. The authors followed recent efforts, in particular Zhu et al. 2022, Zou et al. 2020, etc. to develop a "displacement"-parametrized neural network based kernels in a Hawkes process model to address the potential computational challenges. The empirical results demonstrated the efficacy of their proposed DNSK.

Clearer discussion besides the illustration in Figure 1 may be needed for clearer motivation and insights. For example, it may be necessary to discuss the pros and cons of factorized kernel assumptions and displacement-based parametrization. When introducing Figure 1, the authors may want to briefly mention the computational advantages and difference of the actual event prediction performance before and after using "displacement" on the kernels with the same rank. Although the figure did show that the displacement-based kernel matrix with rank 7 also has three peaks as the original kernel with rank 298, it may be necessary to provide numerical kernel performance measure. The authors may also want to discuss more about the reason for building up the kernels using linear combinations of MLP-based basis functions. As MLPs are assumed to have good approximation capability, is it still necessary to have the linear combination of these MLPs to make the kernel more complicated?

The authors may want to make there empirical performance comparison results more consistent to have all the results reported from all the selected baselines. For example, it may be interesting to explore on the effectiveness of log-likelihood because the potential mismatch of log-likelihood and MSE regarding the THP performance on 1D Data set 2 and 1D Data set 3. For the real-world data, instead of showing the performances of some baselines, the authors may want to present the results from all the applicable baselines.





**Summary Of The Paper:**

The authors of this submission proposed a deep non-stationary kernel (DNSK) modeling Spatio-Temporal Point Processes (STPP) for potentially non-stationary events in continuous time and space.

The authors focused on Hawkes process assuming that the influences from the past events are linearly additive and in turn modeled the conditional intensity function as $\lambda(t, s) = \mu + \sum_{History} k(t, t', s, s')$, where $k(t, t', s, s')$ is named as the influence kernel function, which captures the spatio-temporal dependency. Specifically, DNSK assumes a factorized kernel along time and space directions, as well as potential event "marks" when they are available to handle potential computational challenges. Furthermore, by considering parametrization with "displacement", the authors claimed that DNSK achieves low-rank kernels for more efficiency. The log-barrier method is introduced to preserve non-negativity of the conditional intensity function, also maintaining model interpretability and computational efficiency.

Experiments were performed comparing some existing methods: RMTPP, Neural Hawkes (NH), Transformer Hawkes process (THP), Parametric Hawkes process (PHP), Neural spectral marked point processes (NSMPP) with the proposed DNSK without log-barrier but with a non-negative softplus activation function (DNSK+Softplus). The results showed that the proposed method can indeed capture the non-stationarity and achieve good model recovery and event prediction performances.

**Summary Of The Review:**

The authors proposed DNSK to efficiently model potentially non-stationary events in continuous time and space. Empirical performance comparison results have shown the efficacy of the proposed DNSK methods. Better justification of the adopted kernels, in particular linear combination of MLP-based kernel bases, may be needed. More comprehensive empirical experimental results may need to be provided to confirm the benefits of DNSK compared to the previous similar efforts.

====post-rebuttal====

I truly appreciate the extensive efforts from the authors to address my questions.

---

> ### Author Response · Authors · 2022-11-19
> **Response to Reviewer GLBW**
>
> Thank you for all the insightful reviews, below we answer your questions:
>
> > Clearer discussion besides the illustration in Figure 1 may be needed for clearer motivation and insights. For example, it may be necessary to discuss the pros and cons of factorized kernel assumptions and displacement-based parametrization.
>
> Thank you for the great suggestion. We have now clarified in Section 1 the motivation and benefits of using displacement-based kernel parameterization, in terms of enhancing kernel low-rankness and model representation power.
> We have now provided an additional mathematical argument in Appendix A.1 to show that the kernel SVD-type decomposition is principled, based on Mercer's Theorem and singular value decomposition for general kernel functions. Please refer to Question 1 & 2 in Major Response for more details.
>
> > When introducing Figure 1, the authors may want to briefly mention the computational advantages and differences of the actual event prediction performance before and after using "displacement" on the kernels with the same rank. Although the figure did show that the displacement-based kernel matrix with rank 7 also has three peaks as the original kernel with rank 298, it may be necessary to provide numerical kernel performance measure.
>
> Thank you - we have clarified and presented the computational benefit of our method in Table 1, 2, and Figure 2, comparing the model before using the displacement (corresponding to **NMSPP**) and after using displacement on the kernels (our method) with the same rank, our approach is more efficient (because of low-rank, we need much fewer basis functions) and of better recovery quality.
>
> > The authors may also want to discuss more about the reason for building up the kernels using linear combinations of MLP-based basis functions. As MLPs are assumed to have good approximation capability, is it still necessary to have the linear combination of these MLPs to make the kernel more complicated?
>
> Thanks for your great question. In our case, the kernel function is six-dimension over a continuous spatio-temporal space. Although the MLPs are flexible in terms of function approximation, accurately representing a kernel function over a continuous high-dimensional space would be extremely hard, which requires enormous model and computational resources (like hundreds of layers of MLPs).
>
> The kernel decomposition is mathematically principled, as shown in Appendix A.1. The high-dimensional kernel can be represented by basis functions in low-dimensional space with finite rank, and function approximation in low-dimensional space using MLPs is more accurate and efficient (We only use MLPs with two-hidden layers in our experiments).
>
> > The authors may want to make there empirical performance comparison results more consistent to have all the results reported from all the selected baselines. For the real-world data, instead of showing the performances of some baselines, the authors may want to present the results from all the applicable baselines.
>
> Thank you - we have now implemented additional experiments for *all* baselines across *all* synthetic and real-world data in Section 5. The results show that **DNSK+Barrier** works consistently well by providing higher model log-likelihood and lower predicting errors against other baselines.
>
> > For example, it may be interesting to explore on the effectiveness of log-likelihood because the potential mismatch of log-likelihood and MSE regarding the THP performance on 1D Data set 2 and 1D Data set 3.
>
> Thank you for the great suggestion. After carefully working on it, we have provided more comprehensive comparisons against **THP**: our method is still better than **THP** in achieving both higher likelihood and lower MSE error in Table 2, 3, 4, and Figure 2. A possible reason is that **THP** assumes a linear decaying form of past events’ influence, which we do not need to assume in our model.
>
> > There are also several language problems throughout the submission.
>
> We apologize for the confusion and have now fixed the notation and presenting issues throughout the paper.

---

### Official Review · Reviewer_ZPhD · 2022-10-27

**Confidence:** 3
**Clarity, Quality, Novelty And Reproducibility:** This manuscript is clear and somewhat…
**Correctness:** 3
**Technical Novelty And Significance:** 3
**Empirical Novelty And Significance:** 3
**Recommendation:** 6

**Strength And Weaknesses:**

Strength:
1. The idea of using deep non-stationary kernel in the point process to model spatial-temporal data is interesting and somewhat novel.
2. The method is clear and computational complexity is made.
3. The experiments are sufficient to demonstrate the effectiveness of the proposed method.



Weaknesses

1. The kernel k is designed with heuristics, and can it guarantee that k has good properties, such as to be positive definite?
2. It is said that " (ii) The non-stationarity of events’ influence over spatial-temporal space can be conveniently captured by non-constant psi_l and u_r". I wonder if this is like a kind of positional coding of time index? If so, transformer also has the ability to model non-stationary temporal data, and the experimental results show that THP performs comparable as the proposed methods. The authors did not make a detailed analysis of THP, nor did they compare the differences and advantages of THP with the proposed methods.

**Summary Of The Paper:**

This manuscript proposes a method to model non-stationary spatio-temporal events in the framework of hawkes process. The specific method is to construct a more refined kernel function.

**Summary Of The Review:**

It is an interesting work on modeling non-stationary spatial-temporal data and the proposed method is solid and demonstrated to be effective.

---

> ### Author Response · Authors · 2022-11-19
> **Response to Reviewer ZPhD**
>
> Thank you for all the insightful reviews, below we answer your questions:
>
> > The kernel $k$ is designed with heuristics, and can it guarantee that $k$ has good properties, such as to be positive definite?
>
> Thank you for bringing up the question of the properties of the kernel function. In the revised manuscript, we have added derivation in Appendix A.1 to show that the design of the form of $k$ is principled. The decomposition is of kernel SVD-type, and the technical assumptions require the existence of a low-rank approximation only.
> The properties of the expressed kernel are also clarified in the newly added Remark 1 at the end of Section 3.1. In particular, there is no need for the kernel to be positive definite or even symmetric so that the kernel can express general non-stationary patterns in spatio-temporal point process data.
>
> > It is said that ``(ii) The non-stationarity of events’ influence over spatial-temporal space can be conveniently captured by non-constant $\psi_l$ and $u_r$''. I wonder if this is like a kind of positional coding of time index? If so, transformer also has the ability to model non-stationary temporal data, and the experimental results show that **THP** performs comparable as the proposed methods. The authors did not make a detailed analysis of **THP**, nor did they compare the differences and advantages of **THP** with the proposed methods.
>
> We have now implemented additional experiments on both synthetic and real-world data to compare with the transformer with position coding in Section 5. Detailed analysis has been provided.
>
> While **THP** with positional encoding can capture data non-stationarity, we show that our proposed model **DNSK+Barrier** still outperforms **THP** in Figure 2, Table 2, 3, and 4. A possible reason is that **THP** assumes a linear decaying form of past events' influence, which we do not need to assume in our model.

---

### Official Review · Reviewer_ggHf · 2022-10-30

**Confidence:** 3
**Clarity, Quality, Novelty And Reproducibility:** 1. This paper have a clear presentati…
**Correctness:** 3
**Technical Novelty And Significance:** 3
**Empirical Novelty And Significance:** 1
**Recommendation:** 8

**Strength And Weaknesses:**

Strength:
1. The proposed methods significantly reduces the model complexity without sacrificing the performances. The motivation is clear and the contribution is satisfied.
2. Comprehensive and well designed experiments are conducted.
3. This paper is easy to follow with good presentation.

Weaknesses:
1. Some preliminary knowledge needs to be included to make the paper self-contained. For example, it would be better to briefly describe the "mask point process".
2. Some notations need more clarifications. For example, what "B" refers to in Equation 2. And what the dimension of "s". In Equation 4, there is operation of "s-s'", while "s" refers to the location which should be a 2-D or 3-D value. How the "-" operation is conducted in these values?
3. More discussions on the motivation of specific techniques can be provided. For example, compared to directly using the neural network, what is the advantage of the kernel in Equation 4.
4. No clear problem formulation. For better understanding the problem and methodology, it will be great to have a formal problem formulation.
5. The meaning of "testing l" in Table 4 is not explained.


**Summary Of The Paper:**

This paper proposes a deep non-stationary kernel for spatio-temporal point processes using a different parameterization scheme, which reduces the model complexity. The non-negativity of the solution is guaranteed by a log-barrier method which maintains the linearity of the conditional intensity function. In addition, a computationally efficient strategy for model estimation is introduced. Both the synthetic and real data sets are used to validate the superiority of the proposed model.

**Summary Of The Review:**

The major contribution of this paper is to largely reduce the complexity of the existing neural kernel-based approach towards the spatial-temporal point process. The proposed technique is novel and technical sounds. Though no much improvement on performance and the paper is not solving a new problem, the complexity is largely reduced without sacrificing the performance. Thus, I admit the  contribution and novelty of this paper are above-average. There are some minor issue towards statements and descriptions which have been mentioned above.

---

> ### Author Response · Authors · 2022-11-19
> **Response to Reviewer ggHf**
>
> Thank you for all the insightful reviews, below we answer your questions:
>
> > Some preliminary knowledge needs to be included to make the paper self-contained. For example, it would be better to briefly describe the "mark point process".
>
> Thank you for the great comment. We have now added background knowledge of marked point processes in the paper, and we moved the content related to marked point processes (including background, kernel framework, and efficient model computation) to Appendix B to make the main paper clearer within the page limits.
>
> > Some notations need more clarifications. For example, what $B$ refers to in Equation 2. And what the dimension of $s$. In Equation 4, there is operation of $s-s'$, while $s$ refers to the location which should be a 2-D or 3-D value. How the "-" operation is conducted in these values?
>
> We apologize for the confusion and have now fixed the notation issues throughout the paper.
> $B(s, \Delta s)$ refers to a ball centered at $s \in \mathcal{S} \subset \mathbb{R}^{d_\mathcal{S}}$ with radius $\Delta s$, and $d_{\mathcal{S}}$ refers to the dimension of location space $\mathcal{S}$. The minus between two multi-dimensional values (like $s-s^\prime$ when $s$ is 2D) refers to the element-wise difference between them.
>
> > More discussions on the motivation of specific techniques can be provided. For example, compared to directly using the neural network, what is the advantage of the kernel in Equation 4.
>
> Thank you for the great suggestion. We have now clarified in detail the advantages of using kernels over directly using neural networks, which have been added to the paper. Please refer to Question 1 in Major Response for more details.
>
> We also elaborate on our kernel derivation by providing the mathematical arguments in Appendix A.1, based on the singular value decomposition of general kernel functions and Mercer's Theorem. Please refer to Question 2 in Major Response for more details.
>
> > No clear problem formulation. For better understanding the problem and methodology, it will be great to have a formal problem formulation.
>
> Thank you for the great feedback. We aim to provide efficient modeling with strong representation power for the conditional intensity function in spatio-temporal point processes, which characterizes the spatio-temporal distribution of events. Such distributions are of significant scientific and practical interest both for predicting new events and understanding the events’ generative dynamics. We have also revised Section 1 in the paper to highlight the problem formulation in our work.
>
> > The meaning of "Testing $\ell$" in Table 4 is not explained.
>
> We have now explained ``Testing $\ell$'' in captions of Table 3 and 4, which refers to model log-likelihood on testing data.
>
> > The authors do not provide the source codes and there are no detailed hyper-parameters towards the architecture of the model.
>
> We will post the source codes publicly after the acceptance of the paper. We have added more details about model architectures in Appendix C.

---

> > ### Comment · Reviewer_ggHf · 2022-11-25
> > **Clear response**
> >
> > Thanks for the clarification. I have raised my score.

---

### Official Review · Reviewer_D4xW · 2022-11-01

**Confidence:** 3
**Correctness:** 3
**Technical Novelty And Significance:** 3
**Empirical Novelty And Significance:** 3
**Recommendation:** 6

**Clarity, Quality, Novelty And Reproducibility:**

quality: good

clarity: good

originality: good

**Strength And Weaknesses:**

Strength:
a new kernel method is proposed by considering absolute time
complexity analysis is provided
empirical evaluation performance is strong

Weakness:
additional empirical study would provide more understanding of the method.

Other comments:
- A baseline RNN model with the s and t as an input to RNN, which models time-dependent change in function and similar to your base kernel, should be considered. It would highlight the necessity of using kernel.
- Background does the spatial aspect of the STPP indicate that the change in location always be positive in location? what's an example for such application?
- does the grid resolution have impact on the learning of the kernel?
- would a large t value, corresponding to long term dependency, pose any numerical issue ?
- Eq4: it would be helpful if there is some formal statement on assumptions, the generalization result or its expressivity related to Eq 3.
- Table 3: how does DNSK+softplus compare with transformer results? Would using barrier improve THP as well?


**Summary Of The Paper:**

the paper proposes a more general form of kernel function that is typically used in temporal/spatio-temporal point process, by considering an absolute time-dependent component in addition to the relative spatial-time inputs. In addition, authors made another contribution by proposing a more efficient approach  to ensure the positivity of intensity in the form log-barrier to the optimization problem. Empirical results show both accuracy and efficiency gain over baselines.

**Summary Of The Review:**

novel formulation on the new kernel with promising results.

---

> ### Author Response · Authors · 2022-11-19
> **Response to Reviewer D4xW**
>
> Thank you for all the insightful reviews, below we answer your questions:
>
> > A baseline RNN model with the $s$ and $t$ as an input to RNN, which models time-dependent change in function and similar to your base kernel, should be considered. It would highlight the necessity of using kernel.
>
> We would like to clarify that in Section 5, we have already compared with Recurrent Marked Temporal Point Processes (**RMTPP**), which is precisely the modified RNN for point process data.
>
> We note that **RMTPP** fails to capture the data non-stationarity demonstrated by the experiment results of synthetic data due to their recurrent structures. And the comparisons between our approach and other baselines again highlight the necessity and benefits of using the deep kernel structure to model both data non-stationarity and influence propagation of history flexibly.
>
> > Background does the spatial aspect of the STPP indicate that the change in location always be positive in location? what's an example for such application?
>
> We apologize for the confusion and have revised the background section of spatio-temporal point processes: the change in the location does not need to be positive. The $\Delta s$ in the definition of conditional intensity function refers to the radius of the ball centered at $s$ instead of the change in location.
>
> > Does the grid resolution have impact on the learning of the kernel?
>
> We have now added Appendix A.3 to elaborate on the grid-based model computation and the sensitivity of our model to the choice of grids. The experiment results in Table A.1 reveal that our choice of grid resolution is good enough to capture the complex dynamics of event occurrences for this non-stationary data. The model performance is robust to different grid resolutions.
>
> > Would a large $t$ value, corresponding to long-term dependency, pose any numerical issue?
>
> In our experiments, we observe that setting a larger $t$ (actually, $\tau_{\rm max}$) to capture long-range dependency does not cause numerical issues.
>
> > Eq4: it would be helpful if there is some formal statement on assumptions, the generalization result or its expressivity related to Eq 3.
>
> Thank you for the great suggestion. We have now elaborated on our kernel derivation by providing the mathematical arguments in Appendix A.1 based on the singular value decomposition of general kernel functions and Mercer's Theorem. Please refer to Question 2 in Major Response for more details.
>
> > Table 3: how does **DNSK+softplus** compare with transformer results? Would using barrier improve **THP** as well?
>
> We have added the experimental results in our main paper (See Table 2, 3, and 4), including the performance of **DNSK+Softplus** and other baselines on both synthetic and real-world data. The results show that **DNSK+Softplus** achieves higher data log-likelihood and smaller prediction errors than **THP**.
>
> The barrier method does not impact the performance of **THP** because the softplus activation function already guarantees the non-negative intensity. And how to properly ensure the non-negativity of conditional intensity function while improving the model expressiveness and efficiency remains an exciting question for future work.

---

### Author Response · Authors · 2022-11-19
**Major Response**

### We provide our response to the following major comments on a point-by-point basis:

> Motivation and advantages of using kernel-based methods instead of directly using the neural networks. (Related to R2's Q3)

The advantage of the kernel-based method, compared with directly using neural networks, is that we incorporate necessary statistical structures into the model, which will lead to more interpretable results, and our empirical performance will be better than the directly-using-neural-network approaches (such as widely used RNN or transformer-based models (e.g., **RMTPP**, **NH**, **THP**)). Specifically, we would like to explain that the main advantages of kernel-based methods over directly using neural networks are bi-fold:

(i) Our kernel-based approach will enable us to specifically design the model to characterize the non-stationary point processes by designing the kernel, and capture complicated patterns of influence over continuous space through the displacement-based kernel parameterization. In contrast, the model expressiveness of RNN or attention-based models at non-event times (the time between two adjacent events) is restricted by the assumed parametric form of influence decaying.

(ii) Influence kernels can not only provide predictions for new events, but also be highly interpretable in understanding the process generating the events. By evaluating the kernel at different times and locations, we are able to learn non-stationary influence patterns from past events to the current time $(t, s)$. Such an understanding of event dynamics is of both scientific and practical interest.

> Formal statements of motivation, assumption, and generality of the kernel design in Equation 4 from Equation 3. (Related to R1's Q5, R2's Q3, R3's Q1, R4's Q1 & 3)

We have provided an additional mathematical argument in Appendix A.1 to show that the kernel SVD-type decomposition is principled, based on Mercer's Theorem [1] and singular value decomposition for general kernel functions [2]. The finite-rank decomposition can represent the kernel up to a small residual $\varepsilon > 0$. And we represent basis functions by fully-connected neural networks, taking advantage of their well-known universal approximation power. Thus, our framework is capable of approximating a broad range of general kernel functions.

> More numerical experiments and results of all baselines on synthetic and real-world data. Detailed analysis of the comparison between **THP** and our approach. (Related to R1's Q6, R3's Q2, R4's Q4 & 5)

We have now implemented additional experiments for *all* baselines on *all* synthetic and real-world data in Section 5. While **THP** with positional encoding can capture data non-stationarity, we show that our proposed model **DNSK+Barrier** still outperforms **THP** in Figure 2, Table 2, 3, and 4. A possible reason is that **THP** assumes a linear decaying form of past events' influence, which we do not need to assume in our model.

> Hyper-parameters of model architecture and performance sensitivity to the choice of grids used in model computation. (Related to R1's Q3, R2's Q6)

We have provided more details about the choice of hyper-parameters in Appendix C. We have also added one section in Appendix A.3 to elaborate on the grid-based model efficient computation. Figure A.1 visualizes the procedure of computing the integrals in model log-likelihood. The experiment results in Table A.1 reveal that our choice of grid resolution is good enough to capture the complex dynamics of event occurrences for this non-stationary data. The model performance is robust to different grid resolutions.

---
[1] J Mercer. Functions of positive and negative type, and their connection with the theory of integral equations. Philos. Transactions Royal Soc, 209:4–415, 1909.

[2] Mattes Mollenhauer, Ingmar Schuster, Stefan Klus, and Christof Sch ̈utte. Singular value decomposition of operators on reproducing kernel hilbert spaces. In Advances in Dynamics, Optimization and Computation: A volume dedicated to Michael Dellnitz on the occasion of his 60th birthday, pages 109–131. Springer, 2020.

---

### Author Response · Authors · 2022-11-19
**Paper Update**

We thank all the reviewers for the thoughtful reviews and valuable feedback. We have revised our paper addressing all the questions from the reviewers. The major changes are highlighted in blue and summarized below (R1=D4xW, R2=ggHf, R3=ZPhD, R4=GLBW):

- Following the reviewers' suggestions, we have added one section of arguments in Appendix A.1 to demonstrate the mathematical principle of our kernel design. We have also revised Section 3 in the main paper to highlight the capability of our deep kernel to approximate a large class of complicated influence kernels encountered in practice.

- Following the reviewers' suggestions, we have included additional experimental results of *all* baselines on *all* synthetic and real-world data to demonstrate the consistently superior performance of our approach against other baselines. Detailed analysis of model comparisons has also been provided in Section 5.

- Following R2 and R4's suggestions, we have revised Section 1 (Introduction) about the problem statement and the motivation to employ the displacement-based kernel method to further highlight the contributions of our work.

- Following R1's question, we have added one section of Appendix A.3 to elaborate on the grid-based efficient model computation, including a detailed illustration of integral computation and experiments of model sensitivity to the grid resolutions.

Other minor changes have also been made to fix the notation issues and improve the writing. Below, we first respond to some common questions raised by multiple reviewers in Major Response. And then we reply to each reviewer, respectively.

---

### Decision · Program_Chairs · 2023-01-20

**Decision:**

Accept: poster

**Justification For Why Not Higher Score:**

Perhaps it should be higher--I'm not totally convinced though.

**Justification For Why Not Lower Score:**

Authors were in agreement.

**Metareview: Summary, Strengths And Weaknesses:**

This paper adds to the burgeoning literature on spatiotemporal point process modeling by introducing a new non-stationary influence kernel. Computational efficiency is achieved through a low-rank representation. The reviewers found the presentation clear and well-motivated, and complimented the experiments. There were a number of suggestions made by the reviewers, all of which are very do-able, and I expect the authors to follow them in revising the paper.

**Note From Pc:**

if the above contains the word "oral" or "spotlight" please see: "oral" presentation means -> notable-top-5% and "spotlight" means -> notable-top-25%. As stated in our emails, we are disassociating presentation type from AC recommendations